# Analysis of Endocrine and Inflammatory Markers in Preserved Ratio Impaired Spirometry

**DOI:** 10.3390/medsci12020018

**Published:** 2024-03-27

**Authors:** Fabíola Ramos Jesus, Anna Clara Santiago Moraes, Ingrid Lorena Neves da Silva, Fabine Correia Passos, Cristina Salles, Margarida Célia Lima Costa Neves, Gyselle Chrystina Baccan

**Affiliations:** 1Maternidade Climério de Oliveira (MCO/EBSERH), Universidade Federal da Bahia, Salvador 40055-150, Bahia, Brazil; fabiola.jesus@ebserh.gov.br; 2Departamento de Bioquímica e Biofísica, Instituto de Ciências da Saúde, Universidade Federal da Bahia, Salvador 40110-110, Bahia, Brazil; 3Unidade do Sistema Respiratório, Ambulatório Professor Francisco Magalhães Neto-Hospital Universitário Professor Edgard Santos, Universidade Federal da Bahia, Salvador 40110-200, Bahia, Brazil

**Keywords:** cytokines, lung function, disease pulmonary, COPD, DHEA

## Abstract

Chronic Obstructive Pulmonary Disease (COPD) is a disease of the lungs characterized by chronic airflow obstruction. Individuals with preserved ratio impaired spirometry (PRISm) may be at risk for developing COPD. This study aimed to characterize PRISm and COPD patients in terms of their immune response and endocrine profile to identify differences extending beyond lung function. The participants performed the clinical assessment, pulmonary function test, and blood collection to determine serum hormone levels and concentrations of cytokine. Differences were observed in the nutritional status, lung function, and comorbidity. There were no differences in IL-6, IL-8, IL-10, IL-12, and TNF levels between PRISm and COPD groups. Both PRISm and COPD patients have lower dehydroepiandrosterone (DHEA) and dehydroepiandrosterone sulfate (DHEA-S) levels than controls. Correlation analysis of PRISm and COPD patients revealed positive correlations between serum levels of DHEA-S and DHEA, with forced expiratory volume in 1 second (FEV_1_) and forced vital capacity (FVC), which negatively correlated with IL-8 levels. The results indicated that despite differences in lung function parameters, the PRISm and COPD groups exhibited similarities in endocrine profile alterations. This study represents the first attempt to link endocrine with immune markers and lung function in individuals with PRISm.

## 1. Introduction

Chronic obstructive pulmonary disease (COPD) is a disease of the lungs characterized by phenotypic pulmonary conditions, including chronic bronchitis along with mucus hypersecretion, in addition to the presence of emphysema with enlargement and/or destruction of peripheral airspace walls, causing chronic airflow obstruction [1]. Cigarette smoking is the main risk factor for COPD development [2] Alongside the existence of abnormalities in the airways and/or alveoli, COPD is accompanied by chronic respiratory symptoms, i.e., non-fully reversible airflow limitations identified by a forced expiratory volume in 1 second (FEV_1_/forced vital capacity (FVC) < 0.7) [3].

Early diagnosis of COPD can be beneficial with regard to the course of the disease, reducing the need for extensive use of health services and cost related to medicines [4]. Assessment of pulmonary function can be essential in identifying risk factors for the development of COPD as FEV_1_ percentage values below 86% are considered predictive of progression to COPD [5].

A population at risk of COPD, denominated as preserved ratio impaired spirometry (PRISm), can present as <80% predicted FEV_1_ and ≥0.7 post-bronchodilator FEV_1_/FVC as identified through pulmonary function testing [6,7]. PRISm is considered an independent risk factor for COPD progression [5]. A study found the prevalence of PRISm to be higher in females and younger individuals. In addition, individuals with PRISm exhibited lower tobacco consumption, lower FVC values, and no difference in the risk of acute exacerbations compared to individuals with COPD [8]. PRISm and COPD patients reported similar dyspnea symptoms and current cigarette smoking habits [9]. Although PRISm patients were observed to present structural pulmonary abnormalities, such as emphysema and gas trapping, a study found no differences in airway wall thickness in the patients evaluated, regardless of COPD severity (GOLD 0 vs. 1–4) [10].

Differences in COPD development and the clinical heterogeneity of this disease, such as the presence of emphysema, hypersensitivity, and exacerbation, can be explained by to the presence of different cells and cytokines from the innate and adaptive immune systems in the inflammatory response of the disease [11,12]. Among the cytokines involved in the pathogenesis of COPD, TNF, IL-1β, IL-6, IL-8, and VEGF may be notably elevated. These circulating cytokines may be significantly associated with disease severity and could influence the systemic inflammatory manifestations of COPD [13].

Accelerated aging has been observed in COPD patients [14,15]. In addition to increases in pro-inflammatory cytokine levels, particularly IL-6 and IL-8 [15], which characterize an inflammatory state, alterations in some aging-related hormones have also been reported in patients [16,17]. Decreases in DHEA-S and GH, which occur naturally because of the aging process, have been described as more pronounced in patients with COPD. A study demonstrated prematurity in aging, with estimated average accelerated aging corresponding to 24 years and 13 years due to increased DHEA-S and GH levels, respectively, in COPD patients [18]. In addition to the immunoregulatory functions exerted by these hormones [19], DHEA-S also plays a protective role in the lungs [20,21]. Interestingly, no studies have attempted to investigate relationships between inflammatory and endocrine markers in PRISm patients.

In an attempt to further characterize PRISm patients in comparison to COPD patients, the present study assessed the immune response and endocrine profile to identify differences extending beyond lung function. We also investigated relationships between BMI, cytokine and hormone levels, and lung function.

## 2. Materials and Methods

### 2.1. Study Design and Subjects

The present case-control study enrolled patients diagnosed with COPD, PRISm, and controls, between June 2021 and June 2023 at the Respiratory System Service of the Professor Edgar Santos University Hospital Complex—Federal University of Bahia (HUPES-UFBA). The study protocol was approved by the Institutional Review Board of the same hospital (no. CAAE:28744720.6.0000.0049). All data and collection procedures were performed following the provision of informed consent.

Our study included individuals diagnosed according to the 2023 Global Initiative for Chronic Obstructive Lung Disease (GOLD) who met the criteria for COPD (FEV_1_/FVC < 0.7) or PRISm (<80% predicted FEV_1_ with an FEV_1_/FVC ≥ 0.7) [3]. The GOLD was used to assess the severity of airflow limitations in COPD. All selected individuals were aged between 50 and 80 years, with groups matched based on age (±2 years) and sex. There were a total of 96 individuals, i.e., 64 patients (COPD + PRISm) and 32 participants in the control group. A group of controls with normal lung function was included to compare hormone and inflammatory marker levels.

Controls consisted of subjects with no history of pulmonary disease who had no reported episodes of acute infection in the three months prior to inclusion. Individuals were recruited from the university community and the companions of patients seen at the outpatient service. The exclusion criteria for COPD individuals, as well as those with PRISm, consisted of a history of exacerbated heart disease, use of immunosuppressants, oral corticosteroids or hormone replacement, or any orthopedic, neurological, or cognitive impairment. Any research participants who reported exacerbations of lung disease in the prior two months or were dependent on oxygen were excluded. The same exclusion criteria were applied to members of the control group.

### 2.2. Clinical Parameters

Pulmonary function testing involving spirometry lung volume was applied following the American Thoracic Society guidelines [22]. Spirometric reference values for the Brazilian population were used [23]. Assessment of functional capacity was evaluated via the six-minute walking test (6MWT) according to the protocol established by the American Thoracic Society [24]. Reference equations were used to predict the distance on the 6MWT = 622.461 − (1.846 × Age years) + (61.503 × Gender males = 1; females = 0) [25]. Dyspnea was determined using the modified Medical Research Council questionnaire (mMRC) [26].

Body Mass Index (BMI) was calculated by the ratio of weight (kg) to square height (m^2^), and nutritional status was assessed (BMI < 23 Kg/m^2^ was considered underweight, 23.0 ≤ BMI ≤ 28.0 kg/m^2^ normal weight, >28–30 kg/m^2^ overweight and >30 kg/m^2^ as obese) [27]. Arm muscle circumference (AMC) was measured using a millimeter tape at the midpoint of the non-dominant arm, between the olecranon and acromion [28]. Using mid-arm muscle circumference (MAMC) = AMC − (3.14 × triceps skinfold thickness).

Computed tomography was utilized to evaluate the presence of structural abnormalities, such as emphysema and airway wall thickening in PRISm and COPD patients.

### 2.3. Hormones and Cytokine Quantification

Hormone and cytokine levels were determined in serum samples from peripheral blood collected between 8 a.m. and 10 a.m.; all samples were stored at −20 °C until the time of evaluation. GH concentrations were measured via electrochemiluminescence (Lyphocheck Bio-Rad, Hercules, CA, USA). Serum concentrations of DHEA-S and cortisol were measured using chemiluminescent enzyme immunoassay (ACCESS, Beckman Coulter, Fullerton, CA, USA) and of DHEA using ELISA (Euroimmun, Germany). The minimum detection limits for Cortisol, DHEA, DHEA-S, and GH were 0.4 µg/dL, 0.15 ng/mL, 2 µg/dL, and 0.05 ng/mL, respectively.

The determination of serum cytokine concentrations in COPD, PRISm, and control groups was performed using the cytometric bead array method (BD Pharmingen, San Diego, CA, USA), considering the following detection limits: 2.5, 3.6, 3.3, 1.9, and 3.7, for IL-6, IL-8, IL-10, IL-12, and TNF, respectively. Cytometric fluorescence was quantitatively analyzed on a BD LSRFORTESSA cytometer. Data analysis was performed using FlowJo software version 10.9.

### 2.4. Statistical Analyses

All data were analyzed using GraphPad Prism version 9.0 (GraphPad Software, Boston, MA, USA). We first carried out a descriptive analysis of the levels of clinical, hormonal, and inflammatory markers in the three groups. The normality of data was examined using the Shapiro–Wilk test. Continuous variables non-normally distributed were expressed as medians and interquartile range (IQR), while categorical variables were presented as numerical values (%). The Kruskal–Wallis test was used to identify statistically significant differences in the quantitative variables examined across the three groups, while the chi-squared test was used to compare among categorical variables.

The Mann–Whitney test was used to compare clinical characteristics between COPD and PRISm patients. Spearman’s testing was applied to evaluate correlations between clinical parameters, cytokine, DHEA, and DHEA-S levels. *p*-values < 0.05 were considered statistically significant.

## 3. Results

### 3.1. COPD Patients Exhibit More Malnourished and Less FEV_1_

A total of 96 individuals consented to participate in the study. PRISm and COPD patients were similar in terms of age and height. Those with COPD presented lower FEV_1_ and BMI values than PRISm patients. In COPD, 62% (*n* = 32) patients presented airflow obstruction severity classified as severe or very severe (predicted FEV_1_ < 50%). Smoking history (pack-years) did not differ between COPD (20, 12–48; median and interquartile range, respectively) and PRISm (37, 20–57; median and interquartile range, respectively) patients. MAMC was significantly reduced in the COPD group compared to the PRISm group (Table 1).

### 3.2. Evaluation of Hormones and Inflammatory Markers in PRISm and COPD Patients

DHEA and DHEA-S low serum levels were found in COPD subjects and PRISm patients compared to controls (Figure 1A,B). No significant differences were identified in serum GH (Figure 1C) and cortisol (Figure 1D) levels in COPD and PRISm patients compared to controls.

Figure 2 illustrates higher IL-6 levels in COPD subjects compared to controls, yet no differences were observed regarding IL-8 between the three groups. No significant differences were observed with respect to IL-10, IL-12 or TNF cytokine levels between the PRISm and COPD groups (Figure 3)—no values are shown for the control group in Figure 3 as levels were below the detection limit.

### 3.3. DHEA-S and DHEA Correlates Positively with Lung Function Parameters and IL-8 Correlates Negatively with DHEA in COPD

Table 2 details positive correlations between DHEA-S, DHEA, and FVC, as well as FEV_1_ in PRISm and COPD subjects together. In addition, a positive correlation was also observed between BMI and IL-12 PRISm and COPD subjects together. No associations were identified between cytokine levels (IL-6, IL-8, IL-10, and TNF) and markers of nutrition. In the PRISm group, a significantly negative association between FVC and BMI (*p* < 0.04) was observed. Relationships between the MAMC anthropometric index and FEV_1_, FVC, and FEV_1_/FVC were observed when analyzing PRISm and COPD subjects together.

## 4. Discussion

Although individuals with PRISm do not present spirometric alterations consistent with a diagnosis of COPD in accordance with the GOLD 2023 guidelines [3], our findings identified similar characteristics between these groups of patients, including the presence of mild airway obstruction, dyspnea, smoking history, and alterations on chest imaging.

While malnutrition is prevalent in COPD [29], higher BMI may be a risk factor for restrictive PRISm [30]. A systematic review concluded that obesity might provoke a respiratory restriction that is reflected in the reduced FVC associated with FEV_1_ [31]. In a study investigating the percentage of these predicted values, increased BMI was found to reduce FVC and FEV_1_, with preserved FEV_1_/FCV ratios [32]. The distribution of fat located in the torso or abdominal region could explain the thoracic restriction caused by excess adiposity, limiting diaphragm movement and the accessory muscles involved in respiration [31]. Mild COPD in the presence of obesity also revealed lower FVC values compared with non-obese subjects [33].

In this study, nutritional assessments using BMI and MAMC revealed more malnutrition in subjects with COPD. A paradoxical relationship has been described, in which lower BMI may be associated with worse lung function, whereas higher BMI may be protective against an accelerated decline in lung function [34]. In people at risk of both COPD and PRISm, moderate obesity may indeed be a protective factor in pulmonary function [35]. Herein, around 50% of the individuals with PRISm were overweight or obese. Thus, we hypothesize that in the PRISm population studied, higher BMI may lessen FVC due to repercussions in ventilatory mechanics.

There is a subclassification in PRISm (FEV_1_/FVC ≥ 0.7 and FEV_1_ < 80%) according to the presence (FVC < 80%) or absence (FVC ≥ 80%) of restrictive lung function [30]. Stratification according to the presence or absence of restrictive spirometric findings was not performed in the present study, which may have influenced the similarity in FVC findings between both study groups. Cardiovascular comorbidities have also been associated with PRISm, as high mortality rates due to coronary heart disease have been evidenced in comparison to individuals with obstructive spirometry findings and those with normal spirometry readings [36].

The aging of the immune system provokes changes in inflammatory cytokine production that lead to increased levels of some circulating cytokines, such as IL-6, IL-8, and TNF, which may exacerbate inflammation in COPD patients [37]. According to the severity classification, it was observed that patients with GOLD stage 3 exhibited higher levels of IL-6 and lower levels of IL-10 than patients with GOLD stages 1–2, indicating an imbalance between pro-inflammatory and anti-inflammatory cytokines in severe COPD [38]. There are few studies that assess cytokine levels in PRISm patients. An evaluation of clinical and immunological markers in HIV-infected individuals with pulmonary impairment showed that higher levels of IL-6 were found to be associated with lower FEV_1_ and higher probability of COPD, while higher IL-10 concentrations were associated with lower possibility of PRISm [39]. In our results, the serum concentrations of IL-10, TNF, and IL-12 in control individuals were below the detection limit, which is expected for healthy individuals. Nonetheless, IL-6 and IL-8 cytokines were detectable in all groups, including controls, which is plausible as these cytokines have been described as elevated in older populations.

Studies have found decreased levels of DHEA-S in stable COPD patients [17,40]. Adult women with low DHEA-S levels exhibited lower FEV_1_ and FVC, regardless of smoking history and the use of corticosteroids [20]. DHEA exerts a bronchodilatory effect, which results in the relaxing of smooth muscle through the blockade of calcium channels [41]. Our results indicated a significant decline in DHEA-S in PRISm and COPD compared to age- and sex-matched controls, which may be a determinant of worse lung function, as this hormone was found to be directed related to lung function parameters.

Overall, COPD is associated with reduced levels of DHEA-S and GH hormones, which suggests these markers are relevant to evaluate healthy versus accelerated aging in affected patients [18]. COPD patients with similar BMI had lower blood levels of GH compared to the control group [42]. Increased BMI measurements have been associated with decreased GH secretion [43]. Contrary to findings in the literature, our research did not identify reduced levels of GH in COPD. Furthermore, we observed that the BMI was higher in the control group than COPD and PRISm, which could influence the interpretation of results. It remains unclear whether cortisol may be altered in stable COPD despite the use of inhaled corticosteroids [44]. A longitudinal study described changes in cortisol levels during life, with small decreases seen in early adulthood, stability during middle age, and increased levels with aging [45]. Since the use of inhaled corticosteroids by study participants was not an exclusion criterion, the effects of the hypothalamic–pituitary–adrenal axis were analyzed by performing morning screening of adrenal insufficiency via serum cortisol levels. Our results revealed no differences in serum cortisol among the three groups.

During the aging process, increases in serum concentrations of IL-6 may be attributable to decreased serum levels of DHEA [19]. An in vitro experiment showed that DHEA is capable of inhibiting IL-6 secretion in monocytes. Reductions in DHEA levels can be harmful, especially in the context of chronic inflammatory disease [19]. DHEA plays an immunomodulatory role in regulating cytokine production, including the inhibition of IL-8 secretion [46]. Peripheral muscle atrophy may be accompanied by increased levels of IL-6 and reduced levels of DHEA-S in COPD [16].

Low MAMC, which indicates low muscle mass, independently predisposes older individuals to mortality [47]. In COPD, the loss of muscle mass has been associated with lower FEV_1_ and a higher presence of symptoms [48]. A study investigating COPD in overweight/obese individuals with preserved muscle mass found increased plasma levels of IL-6 and TNF compared to normal-weight subjects, which suggests that the analysis of systemic inflammation using these cytokines may insufficiently estimate diminished muscle mass in this population [49]. Herein, nutritional status was associated with levels of IL-12 in the overall group (COPD + PRISm). Metabolic alterations, such as the presence of obesity or cachexia, may represent two poles related to systemic inflammation [50]. Malnutrition can induce a low-grade systemic inflammatory response with elevated serum levels of TNF, IL-1β, and IL-6 [51]. However, animals with protein malnutrition exhibiting a 20% loss in initial weight revealed decreased phosphorylation of the transcription factor NF-κβ, which decreased IL-1β and IL-12 production following TNF stimulation [52]. Fat mass and fat-free mass are positively related to the plasma levels of inflammatory markers in COPD, which contradicts the theory that malnutrition in COPD is related to increased systemic inflammation. On the other hand, obesity in COPD patients has been associated with increased C-reactive protein [53]. It follows that relationships between nutritional status and the magnitude of systemic inflammatory response in COPD are not fully understood [54].

The lack of longitudinal follow-up in the present study made it impossible to evaluate relationships between BMI, DHEA-S, and worsened lung function over time, which constitutes a limitation. To minimize potential bias and improve the robustness of our results, subject pairing was performed based on demographic characteristics (age and sex). However, this may have limited the representativeness of the COPD group with respect to levels of severity. It is important to emphasize that the prevalence of PRISm has been estimated at around 12% [7]. We acknowledge that our study was unicentric (carried out at an outpatient pneumology clinic with around 289 patients via the outpatient COPD service) and had a small sample size (approximately 9% of the population served by this clinic consented to participate). Nonetheless, the findings presented herein may provide valuable information on immunoendocrine interactions in subjects with PRISm compared to those with COPD, which is largely absent in the data in the literature.

## 5. Conclusions

This study represents the first attempt to link endocrine changes with immune markers and lung function in individuals with PRISm. Despite differences in lung function parameters, both COPD and PRISm groups exhibited similar endocrine and cytokine profile alterations. The changes observed in DHEA and DHEA-S levels may have implications for both the immune response and lung function in affected individuals. Our results additionally present evidence that nutritional markers may influence pulmonary function.

## Figures and Tables

**Figure 1 medsci-12-00018-f001:**
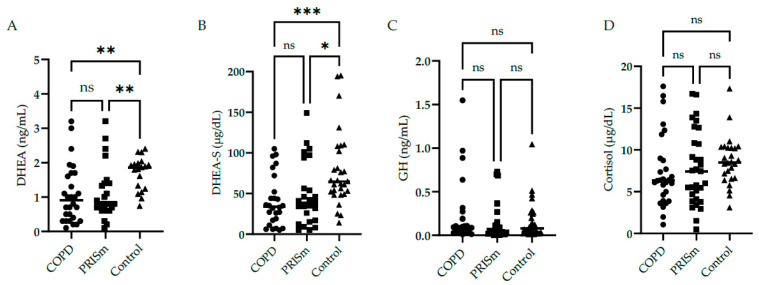
Serum concentrations of DHEA (**A**), DHEA-S (**B**), GH (**C**), and cortisol (**D**) in COPD, PRISm, and control groups. The Kruskal-Wallis test compared COPD (circles), PRISm (squares), and control participants (triangles). Horizontal lines indicate the median of the group. ns = not significant; *** *p* < 0.001; ** *p* < 0.01; * *p* = 0.01.

**Figure 2 medsci-12-00018-f002:**
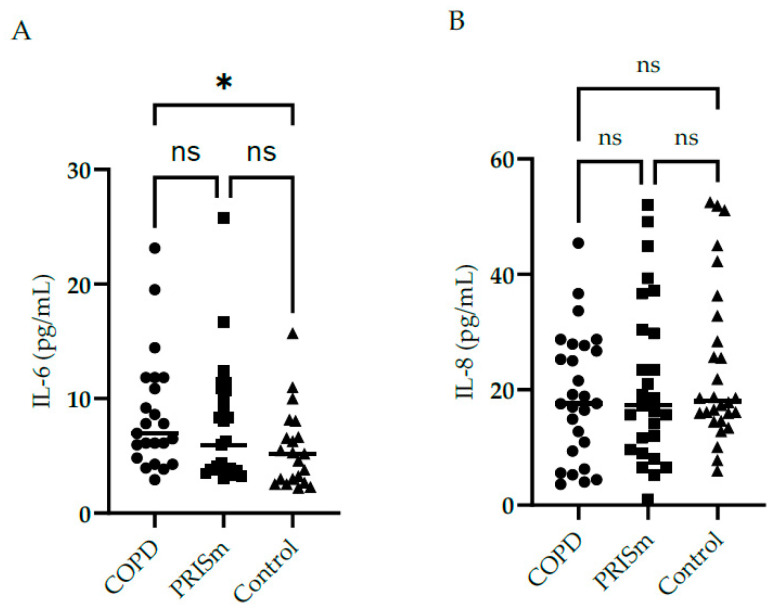
Evaluation of IL-6 (**A**) and IL-8 (**B**) serum concentrations in COPD, PRISm and control groups. The Kruskal-Wallis test compared COPD (circles), PRISm (squares), and control participants (triangles). Horizontal lines indicate the median of the group. ns = not significant; * *p* < 0.05.

**Figure 3 medsci-12-00018-f003:**
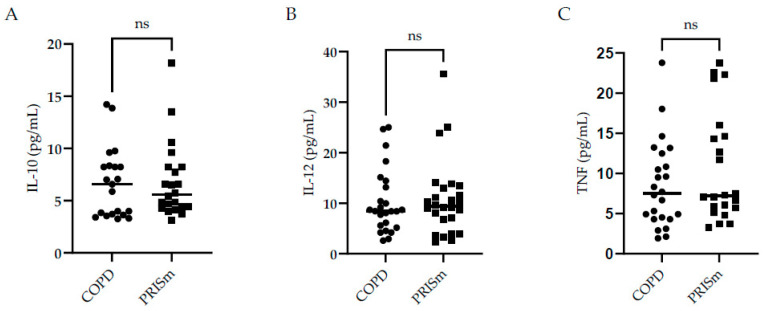
Evaluation of IL-10 (**A**), IL-12 (**B**), and TNF (**C**) serum concentrations in COPD and PRISm patients. TheMann-Whitney test compared COPD (circles) and PRISm participants (squares). Horizontal lines indicate the median of the group. ns = not significant.

**Table 1 medsci-12-00018-t001:** Characteristics and pulmonary function tests of COPD and PRISm patients and controls.

	COPD	PRISm	Control
Male/Female	15/17	15/17	15/17
Age, years; median (IQR)	64 (60–70)	64 (60–72)	63 (59–68)
BMI kg/m^2^; median (IQR)	23 (19–29) ^#^	27 (23–33) *	27 (24–31)
<23 kg/m^2^, *n* (%)	17 (53) ^##^	6 (19) **	4 (13)
23–28 kg/m^2^, *n* (%)	7 (22) ^#^	12 (37)	15 (47)
>28 kg/m^2^, *n* (%)	8 (25)	14 (44)	13 (40)
FVC, L; median (IQR)	1.96 (1.5–2.9) ^#^	1.96 (1.5–3.0) ^#^	2.64 (2.0–3.3)
FEV_1_, L; median (IQR)	0.95 (0.82–1.36) ^###^	1.44 (1.14–2.26) **^#^	2.19 (1.86–3.1)
FEV_1_% pred, (%); median (IQR)	62 (51–76) ^###^	65 (54–72) ^###^	94 (81–100)
FEV_1_/FVC, (%); median (IQR)	50 (40–65) ^###^	73 (71–77) ***^##^	83 (83–94)
Current smoker			
Yes, *n* (%)	12 (38) ^#^	8 (25)	5 (14)
No, *n* (%)	18 (67) ^#^	24 (75)	9 (33)
MAMC, cm; median (IQR)	21 (19–26)	26 (22–27) **	-----
mMRC scale; median (IQR)	2 (1–3)	2 (1–3)	-----
Exacerbations in last year	12 (37.5)	7 (21.8)	-----
6MWTD, % predicted	66 (54–76)	73 (60–80)	-----
Emphysema, *n* (%)	23 (72)	20 (62)	-----
Airway wall thickening, *n* (%)	13 (41)	9 (31)	----
Chronic disease			----
Heart disease, *n* (%)	6 (19)	14 (44) *	----
Diabetes mellitus, *n* (%)	9 (28)	10 (31)	----
Osteoporosis, *n* (%)	2 (6)	5 (16)	----
SAH, *n* (%)	16 (50)	21 (65)	----

Abbreviations: IQR, interquartile range (25th–75th percentile); *n*, number; BMI, body mass index; mMRC, British Council for Modified Medical Research scale; FEV_1_, forced expiratory volume in 1 s; FEV_1_% pred, predicted percentage of FEV in one second; FVC, forced vital capacity; MAMC, mid-arm muscle circumference; SAH, systemic arterial hypertension; 6MWTD, six-minute walk test. * *p* < 0.05 compared to COPD; ** *p* < 0.005 compared to COPD; *** *p* < 0.0001 compared to COPD; ^#^
*p* < 0.05 compared to Controls; ^##^
*p* < 0.005 compared to Controls; ^###^
*p* < 0.0001 compared to Controls.

**Table 2 medsci-12-00018-t002:** Correlations between clinical characteristics, serum cytokine levels and DHEA-S in PRISm and COPD patients.

		FEV_1_, L	FVC, L	FEV_1_ pred, (%)	FEV_1_/FVC, (%)	IL-6	IL-8	IL-10	IL-12	TNF
		r	r	r	r	r	r	r	r	r
COPD	BMI	0.20	−0.01	0.06	0.31	0.32	0.21	0.05	0.30	0.11
	MAMC	0.31	0.22	−0.01	0.14	0.00	-0.17	0.05	0.18	0.06
	DHEA	0.47 *	0.37 *	0.05	−0.18	−0.07	−0.62 **	0.34	0.00	0.26
	DHEA-S	0.49 *	0.38	0.05	−0.12	−0.14	−0.52 **	−0.00	0.11	0.01
PRISm	BMI	−0.33	−0.35 *	0.33	−0.12	0.17	0.01	0.20	0.13	0.20
	MAMC	0.37 *	0.32	0.35 *	0.01	0.04	0.07	0.26	0.25	0.02
	DHEA	0.24	0.23	0.05	0.13	0.03	−0.13	0.08	0.04	0.05
	DHEA-S	0.53 **	0.52 **	0.29	0.09	0.13	−0.01	0.16	0.14	−0.11
COPD+ PRISm	BMI	0.10	−0.14	0.13	0.37 **	0.22	0.12	0.21	0.28 *	0.12
	MAMC	0.40 ***	0.26 *	0.23	0.35 **	0.09	−0.00	0.16	0.32 *	0.08
	DHEA	0.29 *	0.36 **	0.06	0.04	−0.01	−0.39 **	0.24	−0.03	0.22
	DHEA-S	0.42 **	0.52 ****	0.28 *	0.08	0.02	−0.24	−0.05	0.13	−0.01

Abbreviations: BMI, body mass index; MAMC, mid-arm muscle circumference; FEV_1_, forced expiratory volume in 1 s; FEV_1_% pred, FEV_1_ in percent of predicted value; FVC, forced vital capacity. Note: * *p* < 0.05; ** *p* < 0.01; *** *p* < 0.005; **** *p* = 0.0001.

## Data Availability

The data that support the findings of this study are available from the corresponding author upon reasonable request.

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
