# Peer review of "Analysis of Endocrine and Inflammatory Markers in Preserved Ratio Impaired Spirometry"

_medsci, 2024, doi:10.3390/medsci12020018_

Round 1
Reviewer 1 Report
Comments and Suggestions for Authors
Fabíola Ramos Jesus et al. attempted to establish the connection between immunological markers, endocrine function, and lung function in patients diagnosed with preserved ratio impaired spirometry. To comprehend the course of the disease, patient study is crucial. It is appreciated that the authors were able to collect blood samples from 96 patients.
· Authors should provide mean ± standard deviation in the statistics.
· Table 1 should be modified with mean ± standard deviation and add separate column for p-value.
· In all the figures add the title/name to the graphs for better understanding.
· Figure 3a is provided with statistics (ns), but it is not provided in 3b and 3c. Please add statistics to make it uniform.
· Authors should provide CT images of Normal, COPD, and PRISm, which will add weightage to the manuscript.
Comments on the Quality of English Language
In the manuscript, authors should carefully check for a few grammatical and spelling errors.
Author Response
For research article: Correlation of dehydroepiandrosterone with inflammatory cytokine levels and lung function in Preserved Ratio Impaired Spirometry
Title changed to: Analysis of endocrine and inflammatory markers in Preserved Ratio Impaired Spirometry
Thank you very much for taking the time to review this manuscript. Please find the detailed responses below and the corresponding revisions/corrections highlighted/in track changes in the re-submitted files
1-Authors should provide mean ± standard deviation in the statistics.
2-Table 1 should be modified with mean ± standard deviation and add a separate column for the p-value.
Reviewer,
The medians were used as a measure of central tendency, since the variables analyzed in the preliminary normality test (Shapiro-Wilk) were not significant. In Table 1, the p-values are summarized in the table legend, as it would be necessary to add several extra columns to the table due to multiple comparisons between COPD, PRISM, and Controls. To address the reviewer's concerns and for transparency, we have included the results of the normality tests along with a revised table with p-values as supplemental material.
3-In all figures, add the title/name to the graphs for better understanding.
We reintroduced the figures as PPT, since the PNG format did not allow the inclusion of figure titles.
4-Figure 3a is provided with statistics (ns) but is not provided in 3b and 3c. Please add stats to make it even.
We thank the reviewer for this observation and have corrected Figures 3b and 3c accordingly.
5-Authors should provide CT images of Normal, COPD, and PRISm, which will add weight to the manuscript.
For the purposes of our study, an experienced pulmonologist analyzed the CT scan results at the time of clinical evaluation of each research participant. Only two parameters were included (presence of emphysema and airway wall thickening) for the two study groups (controls did not undergo CT), as shown in Table 1. Although we agree with the reviewer's suggestion, i.e., the inclusion of representative images would be illustrative for the reader, this was not the focus of our study and, unfortunately, we do not have access to the actual examinations of the patients, only to the reports generated by the radiologist (who did not participate in the research).

Reviewer 2 Report
Comments and Suggestions for Authors
.- General comments:
This study aimed to characterize PRISm and COPD patients, regarding immune and endocrine to identify differences extending beyond lung function. All the participants performed the clinical assessment, pulmonary function test, and blood collection to determine serum hormone levels and concentrations of cytokine. 96 individuals, 64 patients (COPD + PRISm), and a control group (n=32), age and sex-matched. The differences were observed in the nutritional status, lung function, and comorbidity. The results indicated that despite differences in lung function parameters,PRISm and COPD groups exhibited similarities in endocrine and cytokine profile.
It is a good design and well presented study in terms of results and figures.
Doubts arise, why were analytical parameters and complementary studies such as nutritional ultrasound or DEXA not taken into account in the nutritional assessment?
- Minor comments:
.- Table 2. Perhaps it could be clearer and more visual to read if it were presented in vertical orientation.
.- References. 48 citations are provided, congratulations! However, it should be noted that 18 citations (37%) are recent, that is, five years or less since their publication. Consider, if possible, including a more recent quote.
Reference 21, seems incomplete. Review.
Author Response
For research article: Correlation of dehydroepiandrosterone with inflammatory cytokine levels and lung function in Preserved Ratio Impaired Spirometry
Title changed to: Analysis of endocrine and inflammatory markers in Preserved Ratio Impaired Spirometry
|
Response to Reviewr X Comments
|
Thank you very much for taking the time to review this manuscript. Please find the detailed responses below and the corresponding revisions/corrections highlighted/in track changes in the re-submitted files.
Doubts arise, why were analytical parameters and complementary studies such as nutritional ultrasound or DEXA not taken into account in the nutritional assessment?
Dear Reviewer,
We thank you for your careful reading of our manuscript. Unfortunately, other nutritional testing was not carried out due to our project's budget limitations. We agree with the reviewer, as upon review of our results, it became clear that a deeper analysis of nutritional assessment would be appropriate and lend additional insight to our study.
Minor comments:
.- Table 2. Perhaps it could be clearer and more visual to read if it were presented in vertical orientation
The reviewer raises a valid point, and we initially tried placing Table 2 in a vertical orientation. However, due to the inclusion of 12 variables, it exceeds the page margin, even with reduced formatting. As a result, the only solution we could find was horizontal presentation.
References. 48 citations are provided, congratulations! However, it should be noted that 18 citations (37%) are recent, that is, five years or less since their publication. Consider, if possible, including a more recent quote.
We appreciate this helpful observation and have added 4 additional, up-to-date references in the text.
Reference 21, seems incomplete. Review.
We have added the DOI of the article using Mendeley.
Reviewer 3 Report
Comments and Suggestions for Authors
In the paper: Correlation of dehydroepiandrosterone with inflammatory cytokine levels and lung function in Preserved Ratio Impaired Spirometry, the authors provided evaluation of immune and endocrine markers in relation to COPD patients with particular regard to beginning stage of the disease, PRIS.
The authors found a correlation between hormonal status and spyrometric parameters regardless of the stage of the disease, with no differences in cytokine levels between beginning and progress of the disease, too.
The study is interesting and worth of publishing but the authors need to make certain corrections.
Major revision:
1. The authors should emphasis in the conclusion of the abstract what is the main finding of the study.
2. Introduce full names of all Abb. throughout manuscript
3. Add in subtitle 2.1 the number of enrolled subjects of each category instead of giving it in abstract (it is not necessary info for the Abstract)
4. Indicate in the manuscript what is the novelty of your study, what has not been known before, what is for the first time shown.
5. Fig 1-3 add on the ordinate the name of marker measured (i.e. IL8 pg/mL).
6. Mean values are not visible from the graphs. At least for significantly different markers should be given in the results text mean +- Sd, as well as p value.
7. 184-188 confusing, correct the sentence to give clearly obtained results.
8. In conclusion clearly indicate what is for the first time revealed in this study, what has not been known before.
Minor
1. DHEA abb Abstract missing
2. Line 51-54 Sentence is confusing needs to be corrected.
3. Line 65 inflammaging? 63-66 sentence is confusing, needs correction.
4. 68-72 sentences very confusing, explain fully
5. 145 remove italic letters
6. 162. show the number of severe patients, or %(n) everywhere, unify the text of manuscript
7. Correct lines in the first column of Table 1
8. 205 remove FEV FVC parameters, it is explained in introduction
9. Sentence The present study suffers from some limitations, has no meaning could be removed
Author Response
For research article: Correlation of dehydroepiandrosterone with inflammatory cytokine levels and lung function in Preserved Ratio Impaired Spirometry
Title changed to: Analysis of endocrine and inflammatory markers in Preserved Ratio Impaired Spirometry
|
Response to Reviewer X Comments
|
Thank you very much for taking the time to review this manuscript. Please find the detailed responses below and the corresponding revisions/corrections highlighted/in track changes in the re-submitted files
Major revision:
- The authors should emphasis in the conclusion of the abstract what is the main finding of the study.
We agree with the reviewer’s suggestion and have modified the title of the article to better reflect the findings of our study: “Analysis of lung function and endocrine and inflammatory markers in Preserved Ratio Impaired Spirometry”. Accordingly, in the abstract summary, we highlight that the PRISm and COPD groups exhibited similarities in endocrine profile alterations.
- Introduce full names of all Abb. throughout manuscript
We thank the reviewer for calling this to our attention.
- Add in subtitle 2.1 the number of enrolled subjects of each category instead of giving it in abstract (it is not necessary info for the Abstract)
We have made this alteration as suggested.
- Indicate in the manuscript what is the novelty of your study, what has not been known before, what is for the first time shown.
At the end of the introduction section of the manuscript we state: “Interestingly, no studies have attempted to investigate relationships between inflammatory and endocrine markers in PRISm patients.”
- Fig 1-3 add on the ordinate the name of marker measured (i.e. IL8 pg/mL).
We have added the information as requested to these figures.
- Mean values are not visible from the graphs. At least for significantly different markers should be given in the results text mean +- Sd, as well as p value.
Due to the absence of normality via Shapiro-Wilk testing, we opted to use median values in the figures, with p values described in the figure legends.
- 184-188 confusing, correct the sentence to give clearly obtained results.
We have rewritten this section of the text as follows:
Figure 2 illustrates higher IL-6 levels in COPD subjects compared to controls, yet no differences were observed regarding IL-8 between the three groups. No significant differences were observed with respect to IL-10, IL-12 or TNF cytokine levels between the PRISm and COPD groups (Fig. 3)—no values are shown for the control group in Figure 3 as levels were below the detection limit.
- In conclusion clearly indicate what is for the first time revealed in this study, what has not been known before.
Our study represents the first attempt to link endocrine changes with immune markers and lung function in individuals with PRISm. Our results indicate similarities in endocrine and cytokine profile alterations between PRISm and COPD patients compared to controls.
- DHEA abb Abstract missing
We thank the reviewer for calling this to our attention.
- Line 51-54 Sentence is confusing needs to be corrected.
We have corrected this sentence as follows:
PRISm and COPD patients reported similar dyspnea symptoms and current cigarette smoking habits. [8]. Although PRISm patients were observed to present structural pulmonary abnormalities, such as emphysema and gas trapping, a study found no differences in airway wall thickness in the patients evaluated regardless of COPD severity (GOLD 0 vs. 1–4)
- Line 65 inflammaging? 63-66 sentence is confusing, needs correction.
We have changed the word “inflammaging” to “inflammatory state”.
- 68-72 sentences very confusing, explain fully
We have altered the text to clarify these points as follows:
A study demonstrated prematurity in aging, with estimated average accelerated aging corresponding to 24 years and 13 years due to increased DHEA-S and GH levels, respectively, in COPD patients. In addition to the immunoregulatory functions exerted by these hormones, DHEA-S also plays a protective role in the lungs
- 145 remove italic letters
We thank the reviewer for this observation.
- 162. show the number of severe patients, or %(n) everywhere, unify the text of manuscript
We thank the reviewer for this suggestion.
- 7. Correct lines in the first column of Table 1
We have made this alteration as suggested.
- 205 remove FEV FVC parameters, it is explained in introduction
We agree with the reviewer’s suggestion and have removed this unnecessary detail in the discussion section.
- sentence The present study suffers from some limitations, has no meaning could be removed.
We have rephrased the first sentence as follows in accordance with the reviewer’s suggestion:
The lack of longitudinal follow-up in the present study made it impossible to evaluate relationships between BMI, DHEA-S and worsened lung function over time, which constitutes a limitation.